

# Synergistic and antagonistic effects of immunomodulatory drugs on the action of antifungals against *Candida glabrata* and *Saccharomyces cerevisiae*

Miha Tome[1], Jure Zupan[2,3], Zorica Tomičić[4], Tadeja Matos[1] and Peter Raspor[2,5]

[1] Institute of Microbiology and Immunology, Faculty of Medicine, University of Ljubljana, Ljubljana, Slovenia
[2] Biotechnology, Microbiology, and Food Safety, Department of Food Science and Technology, Biotechnical Faculty, University of Ljubljana, Ljubljana, Slovenia
[3] Lek d.d., Ljubljana, Slovenia
[4] Faculty of Technology, University of Novi Sad, Novi Sad, Serbia
[5] Retired from University of Ljubljana, Ljubljana, Slovenia

Corresponding authors
Miha Tome, miha.tome@gmail.com, miha.tome@mf.uni-lj.si
Peter Raspor, peter.raspor@guest.arnes.si

## ABSTRACT

Candidemia and other forms of invasive fungal infections caused by *Candida glabrata* and to a lesser extent *Saccharomyces cerevisiae* are a serious health problem, especially if their steadily rising resistance to the limited range of antifungal drugs is taken into consideration. Various drug combinations are an attractive solution to the resistance problem, and some drug combinations are already common in the clinical environment due to the nature of diseases or therapies. We tested a few of the common antifungal-immunomodulatory drug combinations and evaluated their effect on selected strains of *C. glabrata* and *S. cerevisiae*. The combinations were performed using the checkerboard microdilution assay and interpreted using the Loewe additivity model and a model based on the Bliss independence criterion. A synergistic interaction was confirmed between calcineurin inhibitors (Fk506 and cyclosporine A) and antifungals (fluconazole, itraconazole, and amphotericin B). A new antagonistic interaction between mycophenolic acid (MPA) and azole antifungals was discovered in non-resistant strains. A possible mechanism that explains this is induction of the Cdr1 efflux pump by MPA in *C. glabrata* ATCC 2001. The Pdr1 regulatory cascade plays a role in overall resistance to fluconazole, but it is not essential for the antagonistic interaction. This was confirmed by the Cg*pdr1*Δ mutant still displaying the antagonistic interaction between the drugs, although at lower concentrations of fluconazole. This antagonism calls into question the use of simultaneous therapy with MPA and azoles in the clinical environment.

## INTRODUCTION

The frequency and associated mortality of candidemia and other forms of invasive candidiasis have not decreased over the past two decades despite the introduction of several extended-spectrum triazole and echinocandin antifungal drugs for use in prophylaxis, empiric therapy, and targeted therapy (*Pfaller & Diekema, 2007*; *Pfaller & Castanheira, 2016*).

*Candida albicans* is the dominant pathogen, but the incidence of invasive infections caused by *Candida glabrata* has been steadily rising (*Pfaller et al., 2012b*; *Pfaller et al., 2014*). The most vulnerable populations include transplant patients, patients with AIDS or cancer, those on immunosuppressive therapy, patients receiving total parenteral nutrition, and premature infants (*Pfaller & Diekema, 2010*; *Whaley & Rogers, 2016*). In certain populations, *C. glabrata* even surpasses *C. albicans* as the leading pathogen; these include patients with hematologic malignancies, diabetes mellitus, and patients with an abdominal source of infection (*Hachem et al., 2008*; *Segireddy et al., 2011*; *Khatib et al., 2016*; *Whaley & Rogers, 2016*). The reasons for the rise of *C. glabrata* infections include the introduction of fluconazole in 1990 and its widespread prophylactic use against fungal infections (*Berrouane, Herwaldt & Pfaller, 1999*), a higher rate of antifungal use and intrinsic or acquired resistance of *C. glabrata* to both fluconazole and echinocandins (*Silva et al., 2012*; *Pfaller et al., 2012a*; *Alexander et al., 2013*; *Pfaller & Castanheira, 2016*; *Colombo, Júnior & Guinea, 2017*), and better identification of non-*albicans* species in the clinic (*Liguori et al., 2009*).

One of the main problems when dealing with *C. glabrata* is its intrinsically low susceptibility to azole antifungals (*Vermitsky & Edlind, 2004*) and its ability to develop resistance to several antifungal drug classes (*Pfaller, 2012*; *Glöckner & Cornely, 2015*). For example, resistance to azole antifungals in clinical isolates is mostly connected to mutations in the gene *PDR1*, which encodes the transcription factor for the pleiotropic drug response (*Vermitsky & Edlind, 2004*). Activating mutations in *PDR1* lead to distinct patterns of altered gene expression among Pdr1 targets, commonly leading to overexpression of efflux pumps that lower the bioavailability of the azoles, thus lowering their effectiveness (*Whaley & Rogers, 2016*). The efflux pumps most commonly associated with azole resistance in *C. glabrata* are the ATP-binding cassette (ABC) transporters Cdr1 (*Sanglard, Ischer & Bille, 2001*), Cdr2/Pdh1 (*Miyazaki et al., 1998*), and Snq2 (*Torelli et al., 2008*). Alternative azole resistance mechanisms include petite mutants with increased expression of *CDR1* and *CDR2* through Pdr1 induction and lower levels of ergosterol intermediates (*Brun et al., 2004*; *Tsai et al., 2006*; *Whaley & Rogers, 2016*). Namely, azoles inhibit Erg11 (lanosterol 14-$\alpha$ demethylase in ergosterol biosynthesis), causing disruption of the membrane and the accumulation of toxic sterol intermediate (*Cowen, 2008*). Documented mechanisms of azole resistance also include Upc2A regulated uptake of exogenous sterols with the Aus1 transporter (*Nakayama et al., 2007*; *Nagi et al., 2011*), and mutations or changes in the expression of target gene *ERG11* and genes involved in sterol intermediate synthesis (*ERG3*, *ERG24*) (*Morio et al., 2012*; *Whaley et al., 2017*). However, *C. glabrata* clinical isolates do not appear to utilize azole resistance mechanisms that involve mutations or changes in the expression of genes in the ergosterol biosynthesis pathway (*Sanguinetti et al., 2005*).

*S. cerevisiae* has resistance mechanisms to azole antifungals similar to those in *C. glabrata* through the increased activity of efflux pumps (Pdr5 homologue of Cdr1) induced by Pdr1 (*Moye-Rowley, 2003*), and the dysfunctional mitochondria of petite mutants (*Kontoyiannis, 2000*). *S. cerevisiae* is not usually associated with pathogenesis; however, instances of *Candida*-like infections (*Aucott et al., 1990*; *Murphy & Kavanagh, 1999*; *Piarroux et al., 1999*; *Wheeler et al., 2003*), often connected with its probiotic variant

*Saccharomyces boulardii* (nom. nud.), have been reported (*Lherm et al., 2002*; *Cassone et al., 2003*; *Enache-Angoulvant & Hennequin, 2005*; *Roy et al., 2017*). Food-oriented *S. cerevisiae* and pathogenic *C. glabrata* therefore present an interesting link for observing the development of various aspects of adaptations to the human host and the mechanisms of evolution in the Saccharomycetaceae (*Wheeler et al., 2003*; *Roetzer, Gabaldón & Schüller, 2011*; *Bolotin-Fukuhara & Fairhead, 2014*). *S. cerevisiae* serves as a model organism, and so its regulatory networks and gene functions are extensively studied. A high degree of homology with *C. glabrata*, therefore, makes it possible to utilize the accumulated knowledge from the model organism and apply it to the pathogen.

Developing new antifungals agents is difficult and costly. Synergistic and additive drug treatments are potential strategies for controlling resistance development and evolution because the administration of multiple drugs may disrupt several mechanisms or processes in the pathogen and thus minimize the selection of resistant strains (*Yeh et al., 2009*; *Bollenbach, 2015*). The combination of antifungals flucytosine (5FC) and amphotericin B (AMB) is recommended by the Infectious Diseases Society of America for *Candida* infections (*Pappas et al., 2015*) due to the high rate of resistance developed during 5FC monotherapy (*Barchiesi et al., 2000*). Other combinations between two antifungals have been put through clinical trials (*Scheven et al., 1992*; *Ghannoum & Elewski, 1999*; *Rex et al., 2003*; *Pachl et al., 2006*), but so far only the 5FC + AMB combination therapy has a clinical role (*Pappas et al., 2015*). Common combinations studied are between a commercial antifungal and a specific inhibitor of a protein of interest; for example, antifungals combined with Hsp90 inhibitors (*Cowen, 2013*; *Veri & Cowen, 2014*) or protein kinase C inhibitors (*LaFayette et al., 2010*). Several promising drug combinations against pathogenic fungi have recently been reviewed (*LaFayette et al., 2010*; *Liu et al., 2014*; *Cui et al., 2015*; *Svetaz et al., 2016*).

Due to the nature of the therapy already employed in the clinical environment, drug combinations are often overlooked with regard to their effect on the pathogens, and treatments can become problematic because of unexpected interactions (*Henry et al., 1999*). Possible side effects and the toxicity of certain drug-drug interactions usually focus on the host, whereas the actual pathogens and their role in these interactions are rarely taken into consideration (*Nett & Andes, 2016*). Simultaneous therapy with several overlapping drugs often occurs in the clinic. Such instances often involve administration of antifungal and immunomodulatory drugs (*Pfaller & Castanheira, 2016*). Many immunomodulatory drugs have conserved targets in fungal pathogens; for example, calcineurin is crucial for the survival during membrane stress (*Cruz et al., 2002*), and calcineurin inhibitors (cyclosporine A (CsA) and Fk506) have antifungal properties (*Steinbach et al., 2007*; *Li et al., 2015*; *Denardi et al., 2015*; *Yu, Chang & Chen, 2015*). Methotrexate (MTX), which blocks folic acid metabolism, also has antifungal properties, as it inhibits ergosterol production in *C. albicans* and makes it more susceptible to azoles (*Navarro-Martínez, Cabezas-Herrera & Rodríguez-López, 2006*). Mycophenolic acid (MPA) targets inosine-5′-monophosphate dehydrogenase, which is a crucial enzyme for the *de novo* synthesis of guanine nucleotides (*Shah & Kharkar, 2015*). MPA has antifungal properties and is synergistic with AMB in *C. albicans* (*Banerjee, Burkard & Panepinto, 2014*).

 

**Table 1  Clinical isolates and their source.**

| Strain | Collection number | Species | Source |
|--------|-------------------|---------|--------|
| Sc1 | ZIM 2558 | *S. cerevisiae* | Throat swab |
| Sc2 | ZIM 2566 | *S. cerevisiae* | Sputum |
| Sc3 | ZIM 2247 | *S. cerevisiae* | Adrenal gland |
| Sc4 | ZIM 2255 | *S. cerevisiae* | Lung of man with immune deficiency syndrome |
| Sc5 | ZIM 2260 | *S. cerevisiae* | Bile tube |
| Sc6 | ZIM 2269 | *S. cerevisiae* | Sorghum beer |
| Cg1 | ZIM 2344 | *C. glabrata* | Urine |
| Cg2 | ZIM 2365 | *C. glabrata* | Sputum |
| Cg3 | ZIM 2369 | *C. glabrata* | Bronchoalveolar lavage |
| Cg4 | ZIM 2382 | *C. glabrata* | Urine taken from a permanent catheter |
| Cg5 | ZIM 2385 | *C. glabrata* | Intestine swab (anus, rectum) |
| Cg6 | ZIM 2389 | *C. glabrata* | Urine |

The study's main goal was to observe whether drug combinations of immunomodulatory and antifungal drugs have any modulatory effects on the selected *C. glabrata* and *S. cerevisiae* isolates. Synergy between calcineurin inhibitors and antifungals was detected. Antagonism between MPA and fluconazole (FLC) in most non-resistant strains was detected as well. This antagonism was unexpected, and therefore the underlying mechanism was briefly investigated. Because FLC resistance is commonly connected to the activation of Pdr1 and subsequent overexpression of the Cdr1 efflux pump, their gene expression was analysed. MPA appears to induce overexpression of the *CDR1* efflux pump, but the central role of Pdr1 is questionable. This was further confirmed in a Cg*pdr1*Δ mutant, in which an antagonistic interaction between FLC and MPA was observed, although at lower concentrations of FLC. This antagonistic interaction between FLC and MPA opens the potential to further explore the underlying mechanism and its impact in the clinical environment.

## MATERIALS & METHODS

### Strains

Five *Saccharomyces cerevisiae*, six *Candida glabrata* clinical isolates and one *S. cerevisiae* non-clinical isolate were selected from the Collection of Industrial Microorganisms (ZIM) at the Biotechnical Faculty, Slovenia (Table 1). These were selected from 96 clinical isolates (40 *S. cerevisiae* and 56 *C. glabrata*; full list of strains is in Supplemental Information 5) and nine non-clinical *S. cerevisiae* isolates based on their minimal inhibitory concentrations (MICs) obtained by the reference method for broth dilution antifungal susceptibility testing of yeasts (CLSI M27-A3) (*CLSI, 2008*). The criterion was to select strains with different MICs to cover several spectra of antifungal resistance. *S. cerevisiae* strain Sc6 from sorghum beer was selected because it displayed high tolerance (16 mg/l) towards fluconazole for a non-clinical isolate. *Candida parapsilosis* ATCC 22019 and *Candida krusei* ATCC 6258 were used as the control strains in the drug susceptibility and checkerboard assays. In addition, to explore the antagonistic mechanism of MPA and azole antifungals, we used *C. glabrata* ATCC 2001 and Cg*pdr1*Δ (CAGL0K10780g Δ::*NAT1*) mutant, provided by Karl Kuchler's

laboratory, Medical University of Vienna, from their deletion library (*Schwarzmüller et al.,* *2014*). Cg*pdr1*Δ is isogenic to *C. glabrata* ATCC 2001.

## Media

Strains were preserved in storage media (10% glycerol, 1% NaCl, and 1% Tween 20) at −80 °C. They were revitalized and routinely grown on yeast peptone dextrose (YPD) agar plates (2% Bacto Peptone, 1% yeast extract, 2% dextrose, and 2% Bacto agar) at 35 °C, regularly sub-cultured before each experiment. Throughout the assays we also used YPD broth (2% Bacto Peptone, 1% yeast extract, 2% dextrose), Sabouraud dextrose agar (3% Sabouraud dextrose from Sigma-Aldrich, 2% Bacto agar), and RPMI (1.04% RPMI-1640 from Sigma-Aldrich, 3.453% morpholinepropanesulfonic acid from Sigma-Aldrich, pH adjusted to pH 7 with 10 M NaOH solution) (*Adams et al., 1998*).

## Drugs

Immunomodulatory drugs used for the screening were methotrexate (MTX), mycophenolic acid (MPA) and its derivate mycophenolate mofetil, cyclosporine A (CsA), and tacrolimus (Fk506). We also included a $\beta$-lactam antibiotic amoxicillin trihydrate (AMX) because it is often administered simultaneously with immunomodulatory agents. Antifungal drugs used for the initial screenings were amphotericin B (AMB), itraconazole (ITC), and fluconazole (FLC). For further exploration of the antagonistic mechanism between MPA and azoles, we used posaconazole (POS), ketoconazole (KCT), and voriconazole (VRC). All of the drugs were obtained from Sigma-Aldrich, except Fk506, which was provided by Acies Bio.

Stock solutions of MPA, MTX, CsA, Fk506, AMB, ITC, POS, KCT, VRC, and CLO were prepared in dimethyl sulfoxide (DMSO; Sigma-Aldrich, St. Louis, MO, USA), whereas AMX and FLC were diluted directly in the medium of choice for the assay. All the final drug concentrations were made in media (RPMI for drug susceptibility and checkerboard assay, YPD for further evaluation of the antagonistic interaction). List of stock solutions is in Supplemental Information 2.

## Checkerboard assay

To determine the susceptibility of the selected strains to these drugs and to observe the effects of the drug combinations on them, we used CLSI M27-A3 checkerboard microdilution assay (*CLSI, 2008*). Briefly, drug dilutions and combinations were prepared in RPMI medium in microtiter plates. Negative (only medium) and positive (strain and medium without drug) controls were included. Strains were grown on Sabouraud agar at 35 °C and, after 24 h, one colony was transferred in 1 ml 0.85% saline solution. Inoculum was prepared by diluting yeast cells in RPMI medium to $3–5 \times 10^3$ cells/ml using the automatic ImageJ-counting technique (*Zupan et al., 2013*). A total of 100 μl of this cell suspension was then transferred to 100 μl of drug suspension prepared in microplates. When the DMSO was used for drug dilution, it comprised <1% of the final test volume in the microtiter well. Tested drug concentrations ranged from 200–2 mg/l for MTX, 400–6.25 mg/l for AMX, 120–2 mg/l for MPA, 400–0.25 mg/l for Fk506, 16–0.125 mg/l for CsA, 256–0.25 mg/l for FLC, 256–0.25 mg/l for ITC, 1–0.004 mg/l for AMB, 16–0.068 mg/l for KCT, 16–0.017 mg/l for VRC, and 16–0.25 mg/l for POS. After incubation at 37 °C

for 24, 48, or 72 h, we measured the optical density at 600 nm ($OD_{600}$) with a microplate reader (Tecan, Männedorf, Switzerland). Background optical densities were subtracted from that of each well. *In vitro* susceptibility and drug combination tests were performed at least in biological triplicates.

When we further investigated the observed mechanism of the antagonism between MPA and additional azole antifungals versus *C. glabrata* ATCC 2001 and Cg*pdr1*Δ mutant, we used a version of the assay described above but replaced SAB and RPMI media with YPD agar plates and broth, respectively. We have confirmed that the antagonistic effect in *C. glabrata* ATCC 2001 is present in both RPMI and YPD media, results are in Supplemental Information 1.

## Fractional inhibitory concentrations index

The data obtained from the checkerboard microdilution assays were analysed with the fractional inhibitory concentrations index (FICI) based on the Loewe additivity model (*Loewe, 1928*), using the following equation: $FICI = FIC_A + FIC_B$, where $FIC = MIC_{combination}/MIC_{individual}$. For azoles and immunomodulatory drugs MIC50 was used, and MIC90 for AMB. FICI is interpreted as synergistic when $\leq 0.5$, indifferent when $>0.5$ and $<4$, and antagonistic when $\geq 4$ (*Odds, 2003*). For calculation of the FICIs when the MIC resulted in an off-scale value, the next higher concentration (e.g., $>32 = 64$ mg/l) was used (*Moody, 2010*). $FICI_{min}$ was reported as the FICI in all cases unless the $FICI_{max}$ was greater than 4, in which case $FICI_{max}$ was reported as the FICI for that particular data set (*Meletiadis et al., 2005*).

## Bliss independence

The expected effect of drug combinations was also calculated by a model based on the Bliss independence (BI) criterion, where we assume that the relative effect of a drug at a particular concentration is independent of the presence of the other drug (*Bliss, 1939*; *Goldoni & Johansson, 2007*; *Yeh et al., 2009*). We calculated the predicted decrease of relative growth ($E_{predicted}$) using the following equation: $E_{predicted} = 1 - E_A * E_B$, where $E_A$ and $E_B$ are individually measured relative growth inhibitions by drugs *A* or *B*, respectively. Positive or negative deviations ($\Delta E = E_{measured} - E_{predicted}$) from this predicted decrease of relative growth describe synergistic and antagonistic interactions, respectively (*Yeh et al., 2009*).

To interpret and summarize the entire interaction surface calculated by the BI criterion among several different drug combination concentrations, we used previously described interpretations (*Meletiadis et al., 2005*). Briefly, we summed all statistically significant $\Delta E$ ($\Sigma SSI$), determined the mean percentage (MSSI), and calculated the 95% confidence interval (CI). If it did not include 0 and was positive or negative, statistically significant synergy or antagonism, respectively, was claimed for the entire data set. In addition, we also calculated the $\Sigma SSI$ and MSSI for all the significant synergistic ($\Sigma SYN$ and MSYN, respectively) and antagonistic ($\Sigma ANT$ and MANT, respectively) $\Delta E$ separately. The absolute sum of all $\Sigma SYN$ or $\Sigma ANT$ was considered to be a weak (0–100%), moderate (100–200%), or strong (>200%) interaction.

The interaction between two drugs was considered significant, if it was confirmed with at least one model (either BI or FICI).

### Gene expression

*C. glabrata* ATCC 2001 was grown in liquid YPD overnight at 37 °C. In the morning, we diluted it in fresh YPD to $OD_{600}$ 0.05 and let it grow at 37 °C back to $OD_{600}$ 0.1 (approximately 1.5 h). At that point, we added the following drug combinations: (a) untreated, (b) 5 mg/l FLC, (c) 5 mg/l MPA, and (d) 5 mg/l FLC and 5 mg/l MPA. The optimal concentrations were determined with preliminary growth tests and checkerboard assays. All of the samples received the same amount of DMSO.

At timepoints 0, 2, and 4 h we collected the samples with a 3 min spin down at 1,500 g, 4 °C, resuspended them in ice-cold water, and transferred them to 2 ml screw caps, where we performed a short spin down to remove the supernatant and froze the pellet in liquid nitrogen. Samples were stored at −80 °C. Time points were determined according to growth curve assays (Supplemental Information 4).

RNA isolation and qPCR analysis were performed as described previously (*Tscherner et al., 2012*). *PGK1* and *RIP1* were used as housekeeping genes (*Hnisz et al., 2010*; *Li, Skinner & Bennett, 2012*).

The primers used were CgPGK1f (5′-ACGAAGTTGTCAAGTCCTCCA-3′), CgPGK1r (5′-TTACCTTCCAACAATTCCAAGGAG-3′), CgRIP1f (5′-CTTCATGGTCGGTTCTCT AGG-3′), CgRIP1r (5′-ACAACAACGTTCTTACCCTCAG-3′), CgPDR1f (5′-TACCAATG TCTCAGATACCACCA-3′), CgPDR1r (5′-CTGTCTTTAGAATCCAACTGCGT-3′), CgCDR1f (5′-AGACTTACGCTAGACATTTAACGG-3′), and CgCDR1r (5′-CACAAATA GAGACTTCAGCAATGG-3′). Amplification curves were analysed using the Realplex Software (Eppendorf) and relative mRNA quantification was performed using the efficiency corrected $\Delta\Delta$Ct method (*Pfaffl, 2001*). Quantification was performed in Excel (Microsoft) and statistical analysis with GraphPad Prism (GraphPad Software; GraphPad, San Diego, CA, USA) using one-way ANOVA and Bonferroni's multiple comparison test.

## RESULTS

### Drug susceptibility

Each test had MICs of the control strains (*C. parapsilosis* ATCC 22019, *C. krusei* ATCC 6258) within the expected range for the tested antifungal. MICs for selected strains against individual drugs at 48 h are summarized in Table 2. Strains showed resistance to certain antifungals; for example, Sc2 (32–64 mg/l), Cg5 (64–128 mg/l), and Cg6 (128 mg/l) against FLC and Sc1 (2 mg/l), Sc2 (2–4 mg/l), Sc3 (2–4 mg/l), Cg5 (4–16 mg/l), and Cg6 (128 mg/l) against ITC (*CLSI, 2008*). For most of the immunomodulatory drugs, the concentration range that was used here, and was based on the range expected in human blood after drug administration, did not obtain a MIC; exceptions were some strains with MPA (Sc3, Sc4, Sc5, Cg2 at 120 mg/l) and Fk506 (Sc4, Sc6 at 200 mg/l).

### Interpretation of drug interactions

Results were obtained using the checkerboard microdilution assay. A summary of the interpretations for each strain and drug combination is found in Fig. 1. The interaction was considered significant, if it was confirmed with at least one model (FICI or BI). Calculated FICI and BI values are located in Supplemental Information 3.

**Table 2** Susceptibility of clinical isolates against individual drugs.

| Strain | MIC range (mg/l) | | | | | | | |
|--------|------|------|------|------|------|------|------|------|
| | FLC | ITC | AMB | MPA | MTX | CsA | Fk506 | AMX |
| Sc1 | 4–8 | 1–2[b] | 0.125–0.5 | >120 | >200 | >16 | >400 | >400 |
| Sc2 | 32–64[a] | 2–4[a] | 0.25 | >120 | >200 | >16 | >400 | >400 |
| Sc3 | 16–32[b] | 2–4[a] | 0.25 | 120 | >200 | >16 | >400 | >400 |
| Sc4 | 1–2 | 0.5–1 | 0.5 | 120 | >200 | >16 | 200 | >400 |
| Sc5 | 4–8 | 1–2[b] | 0.25 | 120 | >200 | >16 | >400 | >400 |
| Sc6 | 8–16[b] | 1 | 0.125–0.25 | >120 | >200 | >16 | 200 | >400 |
| Cg1 | 2–4 | 1 | 0.25–0.5 | >120 | >200 | >16 | >400 | >400 |
| Cg2 | 2–4 | 0.5–1 | 0.5–1[b] | 120 | >200 | >16 | >400 | >400 |
| Cg3 | 4–16[b] | 0.125–1 | 0.125–0.5 | >120 | >200 | >16 | >400 | >400 |
| Cg4 | 2–4 | 0.25–0.5 | 0.125–0.5 | >120 | >200 | >16 | >400 | >400 |
| Cg5 | 64–128[a] | 4–16[a] | 0.5 | >120 | >200 | >16 | >400 | >400 |
| Cg6 | 128[a] | 128[a] | 0.125–0.25 | >120 | >200 | >16 | >400 | >400 |

**Notes.**
[a] Resistant.
[b] Susceptible dose dependent according to *CLSI (2008)*.

The FLC + MPA combination was antagonistic in eight out of 12 strains (five *S. cerevisiae*, three *C. glabrata*) and synergistic in two *C. glabrata* strains. One synergistic and two indifferent interactions involved highly resistant strains (MIC of 64 mg/l or higher). The AMB + MPA combination had a synergistic effect in eight out of 12 strains (three *S. cerevisiae*, five *C. glabrata*) and antagonism in one *S. cerevisiae* isolate. The effect of ITC + MPA was not as uniform; four synergistic and three antagonistic interactions were observed, indicating that the response is strain-specific.

A strain-specific response was observed in the combination of antifungals and MTX or AMX as well. MTX + FLC had two synergistic and three antagonistic interactions, MTX + ITC four synergistic and one antagonistic, MTX + AMB two synergistic and five antagonistic, AMX + FLC three synergistic interactions, AMX + ITC two synergistic and one antagonistic, and AMX + AMB five synergistic and four antagonistic interactions.

Out of 72 interactions between antifungals and calcineurin inhibitors (CsA and Fk506), 67 were synergistic, even in the FLC and ITC resistant strains Cg5 and Cg6. Antagonistic interactions were observed only in the combination of CsA and FLC in three *S. cerevisiae* isolates.

The results from the FICI and BI models fitted well, and both showed a trend towards the same interpretation; if the interpretation with one model was synergy, the other model either showed synergy or indifference, but never the opposite, antagonism. Out of 180 combinations tested against selected strains, the FICI model showed 60 significant modulatory interactions and the BI model 124, in which most of the absolute sum values (ΣSYN or ΣANT) indicated strong interactions (>200%).

### Antagonistic interaction between MPA + azole antifungals

The strain *C. glabrata* ATCC 2001 was used to further explore the observed antagonism between MPA and azole antifungals (Fig. 2A). Azole antifungals included FLC, ITC, KCT,

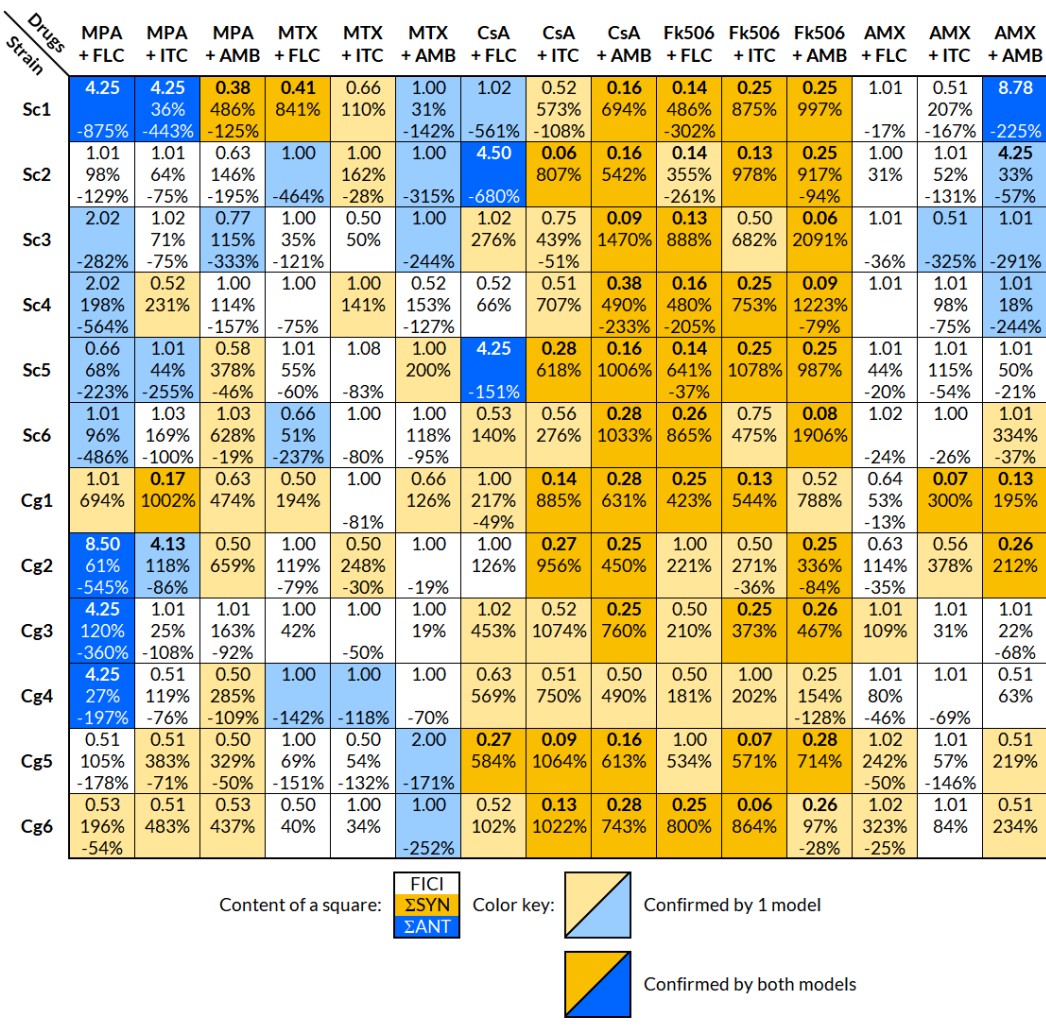

**Figure 1** **Interpretation of the effects of drug interactions against each selected strain with FICI and BI.** Each small square includes the calculated overall FICI and BI values of several experiments combined for each drug combination and strain; list of strains in Table 1. Yellow indicates synergy and blue antagonism. Darker colors signify confirmation of the effect by both FICI and BI, and a lighter color only by one model. The interaction was considered significant, if it was confirmed with at least one model. Interpretation of FICI: synergistic $\leq$ 0.5, indifferent 0.5–4, antagonistic $\geq$ 4 (Odds, 2003). $FICI_{min}$ was reported unless the value of $FICI_{max}$ was greater than 4, in which case $FICI_{max}$ was reported. Interpretation of BI ($\Sigma SYN$ and $\Sigma ANT$): positive values are interpreted as synergy and negative as antagonism, where the absolute sum of all $\Sigma SYN$ or $\Sigma ANT$ was considered to be a weak (0–100%), moderate (100–200%), or strong (> 200%) interaction (Meletiadis et al., 2005). **FICI**, fractional inhibitory concentration index; **BI**, Bliss independence; $\Sigma SYN$, sum of all significant positive values calculated by the BI model; $\Sigma ANT$, sum of all significant negative values calculated by the BI model.

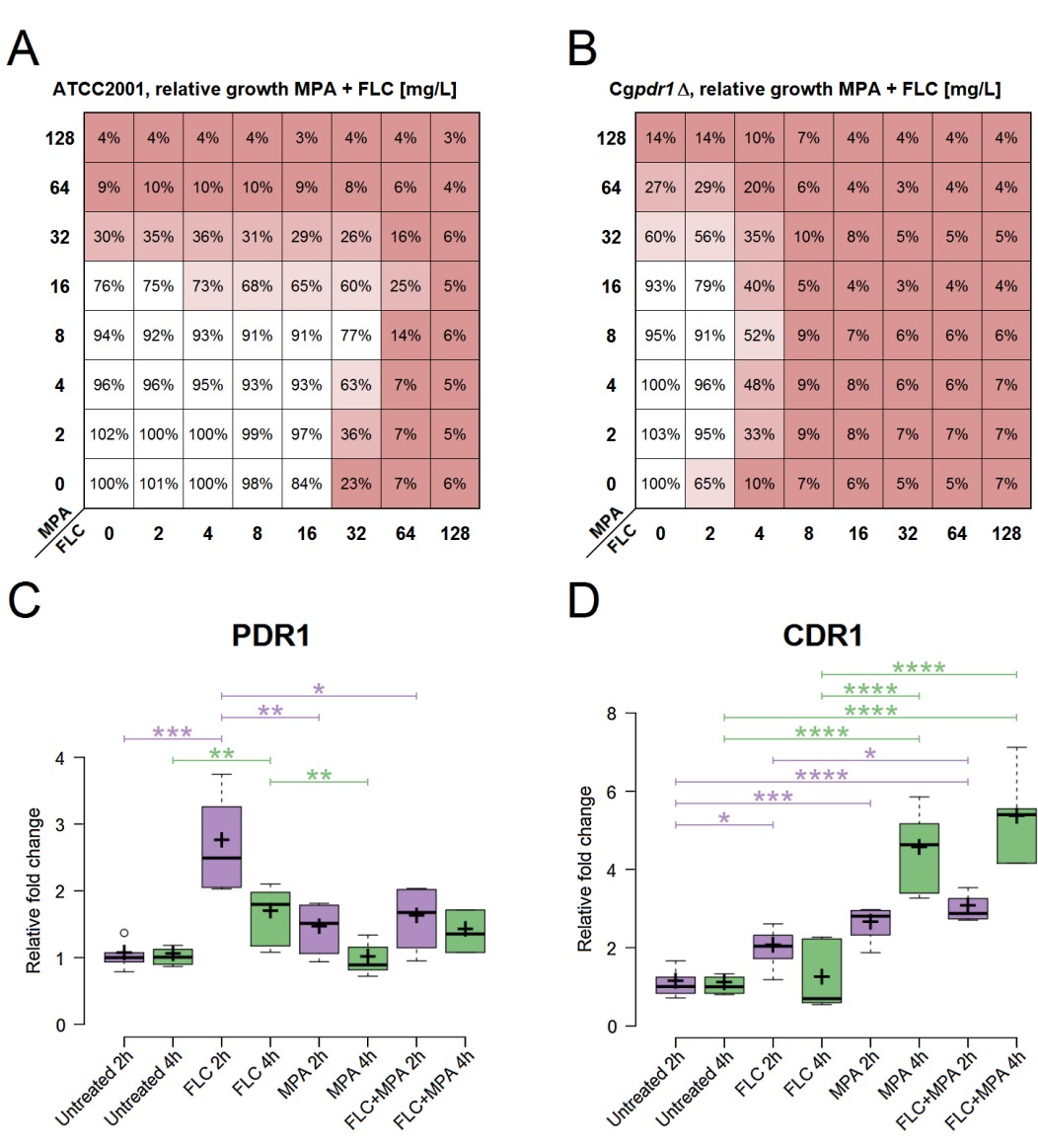

**Figure 2  Role assessment of *PDR1* and *CDR1* for the antagonism in the drug combination of FLC + MPA against *C. glabrata* ATCC 2001.** (A) Antagonistic interaction (interpreted by the BI model with ΣANT of −266.9%) observed via relative growth in a checkerboard assay with *C. glabrata* ATCC 2001 against various combinations of MPA and FLC in YPD at 37 °C. Red indicates lower growth. (B) Relative growth in a checkerboard assay with Cg*pdr1*Δ against various combinations of MPA and FLC in YPD at 37 °C. Antagonism is visible (interpreted by the BI model with ΣANT of −320.79%), most emphasized at MPA 8 mg/l and FLC 4 mg/l. (C, D) Relative fold change of gene expression for *PDR1*, *CDR1*, respectively, in *C. glabrata* ATCC 2001 after 2 and 4 h of four different treatments at 37 °C: YPD broth (untreated) and YPD broth with either 5 mg/l FLC, 5 mg/l MPA, or 5 mg/l FLC + 5 mg/l MPA. The figures show one representative experiment with three independent biological replicates for each strain tested. Statistical analysis was performed in GraphPad Prism using one-way ANOVA and Bonferroni's multiple comparison test. *P*-value of less than 0.05 was significant, statistically significant differences are marked with ∗.

**Table 3** Interactions of azole antifungals and MPA against *C. glabrata* ATCC 2001 interpreted by FICI and BI.

| Drug | MIC (mg/l) | FICI (+ MPA) | | BI (+ MPA) | | |
|------|-----------|--------------|-----|------------|-----------|-----|
| | | (Range) | INT | ΣSYN (*n*) | ΣANT (*n*) | INT |
| FLC | 32 | 1.06–2.5 | IND | 6.42% (1) | −266.9% (15) | **ANT** |
| ITC | 0.25 | 1.25–4.25 | **ANT** | 16.59% (2) | −477.41% (22) | **ANT** |
| KCT | 0.5 | 1.25–4.125 | **ANT** | 0 | −229.8% (12) | **ANT** |
| VRC | 0.25 | 1.063–2.5 | IND | 53.1% (7) | −284.2% (11) | **ANT** |
| POS | 0.25 | 1.5–4.125 | **ANT** | 0 | −321.1% (9) | **ANT** |

Notes.

INT, interpretation; IND, indifference; ANT, antagonism.

*n*, number of significant interactions (out of 49, which combine the entire interaction surface of a single experiment) in one representative experiment with three independent biological replicates for each drug combination tested.

VRC, and POS. Table 3 shows the MIC, FICI, and BI of this strain against azole antifungals combined with MPA (MIC for MPA ranges from 32 to 64 mg/l) in YPD at 37 °C after 48 h. All of the interactions were interpreted as antagonistic by at least one model. This confirms that an antagonistic interaction between MPA and all of the selected azole antifungals occurs in *C. glabrata* ATCC 2001.

## The roles of *PDR1* and *CDR1*

Gene expression of *PDR1* and *CDR1* was analysed in *C. glabrata* ATCC 2001 at two different timepoints (Fig. 2). It was calculated relative to the untreated samples for each timepoint. All of the following differences described have statistical significance ($p$-value $< 0.05$).

At 2 h, *PDR1* expression was significantly higher in samples treated with FLC. It was higher than in all the other conditions, including the samples treated with the combination of FLC + MPA. At 4 h the expression of *PDR1* was still higher in samples treated with FLC compared to the untreated samples and samples treated with MPA.

For *CDR1*, at 2 h, all treated samples (FLC, MPA, FLC + MPA) had higher expression than the untreated samples. At 2 h FLC + MPA had higher expression of *CDR1* than the FLC-treated samples. At 4 h the expression of *CDR1* was even higher in MPA and FLC + MPA-treated samples and was significantly higher than in the samples treated only with FLC.

The expression patterns for *CDR1*, which seemed like a good candidate to explain the antagonism of the drug combination, and the low expression of *PDR1* in the MPA and FLC + MPA-treated samples questioned whether the *PDR1* regulatory cascade has a central role in the drug response mechanism responsible for antagonism between FLC and MPA. To test this, we performed a checkerboard assay with Cg*pdr1* Δ mutant against the combination of FLC and MPA. Figure 2B shows the relative growth from this assay, in which we saw increased susceptibility to FLC (MIC at 4 mg/l) and the antagonistic pattern, which was confirmed by the BI model (ΣANT = −320.79%). These results suggest that the typical *PDR1* drug response is not the only pathway required for this antagonistic interaction and that the induction of *CDR1* in this case could to be regulated by alternative pathways.

## DISCUSSION

Drug combinations can present an attractive way to deal with antifungal resistance and the lack of antifungals, but there may be dangerous complications when there are adverse reactions in the host or antagonism between the drugs. In this study, the combination of MPA and FLC made eight out of twelve strains (five *S. cerevisiae*, three *C. glabrata*; all are less resistant strains) more tolerant to the antifungal. This could have consequences in the clinical environment, and it also opens a path to explore drug resistance mechanisms.

### Synergism: calcineurin inhibitors + antifungals

Certain patterns were observed with different drug combinations (Fig. 1). The most striking one was a generally synergistic effect between most antifungals and calcineurin inhibitors (CsA and Fk506) against all strains tested. The synergistic effect of calcineurin inhibitors and antifungals is well documented (*Cruz et al., 2002*; *Li et al., 2015*; *Denardi et al., 2015*; *Yu, Chang & Chen, 2015*). Our results differ to those of (*Cruz et al., 2002*) who reported synergistic toxicity toward other fungal species but not *S. cerevisiae*. This may be due to a strain-specific response in *S. cerevisiae*, but further tests are required to clarify this.

### Antagonism: MPA + azoles

Eight out of twelve antagonistic interactions between MPA and FLC were found. Antagonism was not observed in strains that had high resistance to FLC (Sc2 32–64 mg/l, Cg5 64–128 mg/l, and Cg6 128 mg/l), which indicates that the antagonism is probably the result of similar mechanisms that produce the high resistance. FLC resistance mechanisms include overexpression of the efflux pumps in most cases (*Whaley & Rogers, 2016*). The antifungal activity of MPA, which is due to the depletion of purine nucleotides (*Banerjee, Burkard & Panepinto, 2014*), made the antagonistic effect all the more surprising. We therefore explored whether the most common mechanism of azole resistance in *C. glabrata* and *S. cerevisiae* (Pdr1 induction of efflux pump Cdr1/Pdr5) had a role in this antagonistic interaction.

### Antagonism: the roles of *PDR1* and *CDR1*

The central role of the transcriptional factor Pdr1 in azole resistance has been described in detail (*Caudle et al., 2011*; *Yibmantasiri et al., 2014*). It is usually linked to the induction of efflux pumps (Cdr1 as a dominant example) that remove the azole antifungals from the cell. In this study, relatively high expression values of *CDR1* were observed in the drug combination FLC + MPA and MPA alone compared to the untreated samples and even to the FLC-treated ones (Fig. 2D). This makes the induction of efflux pumps a good explanation for the increased resistance to azole antifungals. However, an interesting aspect arose when examining the expression values of *PDR1* (Fig. 2C) because their expression pattern did not match the *CDR1* induction. Normally, Pdr1 positively regulates the expression of *CDR1*, but this was not seen here. This suggests that Pdr1 is not the only regulatory mechanism enabling a higher expression of *CDR1* with MPA or the FLC + MPA combination. This was further demonstrated by an antagonistic effect in the Cg*pdr1*Δ mutant, interpreted by the BI model with a ΣANT value of −320.79%, although at lower

concentrations of FLC ([Fig. 2B]). Undoubtedly Pdr1 still plays a major role in overall resistance (because resistance to FLC did drop in the combination versus Cg*pdr1*Δ), but the antagonistic pattern still existed. This opens up new and interesting questions regarding which mechanisms instead of the Pdr1 regulatory cascade are responsible for higher *CDR1* expression and the observed antagonistic effect. Alternative regulators of *CDR1* could include transcriptional factors associated with multidrug resistance (e.g., *RDR1*, *YRR1*, *YRM1*, and *STB5*) or *CAD1*, *MSN2*, and *MSN4* for general stress response, *YAP1* for oxidative stress, *CRZ1* from calcineurin-mediated stress response, or *ECM22* and *UPC2* for cell membrane composition (*Monteiro et al., 2017*; *Teixeira et al., 2018*). In addition to Cdr1, other efflux pumps associated with azole resistance should also be considered in the future, such as Cdr2 (*Miyazaki et al., 1998*), Snq2 (*Torelli et al., 2008*), Flr1 (*Alarco et al., 1997*), Qdr2 (*Costa et al., 2013*), Tpo1_2 (*Pais et al., 2016*), Ybt1 (*Tsai et al., 2010*), Yhk8 (*Barker, Pearson & Rogers, 2003*) and Yor1 (*Vermitsky et al., 2006*). Further studies at the systemic level would be required to obtain a better picture of the entire mechanism.

### Antagonism: clinical environment

There are clinical implications to this discovery. MPA (and its prodrug mycophenolate mofetil) is widely used as a maintenance immunosuppressive regimen in solid organ transplant patients, for the prophylaxis and treatment of acute and chronic graft-versus-host disease, and to promote engraftment after hematopoietic stem cell transplantation (*Zhang & Chow, 2016*). There is no data on the frequency of clinical usage for the combination of FLC + MPA; however, antifungal prophylaxis with azoles (VRC, POS, ITC, and FLC) is commonly prescribed in immunocompromised populations, which also involve the use of MPA (*Pappas & Silveira, 2009*; *Brizendine, Vishin & Baddley, 2011*; *Groll et al., 2014*). Possible antifungal prophylaxis or actual treatments with azole antifungals in these cases should therefore be used with caution and considered for each individual case because induced resistance could result in a failed therapy. There is even a report of a statistically significant increase in fungal infections in the geriatric renal transplant population when receiving MPA versus azathioprine, but the specific organisms and sites of infection were not reported (*Meier-Kriesche et al., 1999*; *Ritter & Pirofski, 2009*). It can be speculated that this antagonism could be connected to this increase in fungal infections, because FLC and other azoles (VRC, POS and ITR) are used for the prophylaxis in solid organ transplantations and other therapies involving immunocompromised patients (*Pappas & Silveira, 2009*; *Brizendine, Vishin & Baddley, 2011*; *Groll et al., 2014*; *Vazquez, 2016*). The next steps should expand the number of tested strains and species against the combination of azoles and MPA, include an *in vivo* evaluation of the combination, and include other drugs involved in a certain therapy as well; for example, a typical cocktail of prednisolone, MPA, and cyclosporine in solid organ transplantations (*Sollinger, 1995*).

## CONCLUSION

This study examined how combinations of immunomodulatory and antifungal drugs affect selected strains of *C. glabrata* and *S. cerevisiae*. It confirmed a strong synergistic toxicity between calcineurin inhibitors and antifungals, but also discovered an antagonistic

interaction between MPA and azoles in non-resistant strains. Based on observation of gene expression in *C. glabrata* ATCC 2001 this is probably due to increased expression of drug efflux pump Cdr1 by MPA. However, the mechanism of the induction is still unknown because the Pdr1 regulatory cascade was not essential for the antagonistic interaction, and this deserves further investigation. In addition, the combined use of MPA and azoles in the clinical environment should be carefully reevaluated. In particular, there is a need to recognize that drug combinations affect not only the host but also the pathogen.

## ACKNOWLEDGEMENTS

We thank Neža Čadež, the curator of the Collection of Industrial Microorganisms (ZIM) at the Biotechnical Faculty, Slovenia, for providing us with the yeast strains; and Karl Kuchler from the Medical University of Vienna and his team of researchers for helping with the technical part of the qPCR analysis and for providing the Cg*pdr1*Δ mutant. We also thank the company Acies Bio, which provided the drug Fk506.

### Funding

Grant number 36377 for junior researcher Miha Tome was financed by the Slovenian Research Agency (ARRS) from the Slovenian national budget. The funders had no role in study design, data collection and analysis, decision to publish, or preparation of the manuscript.

### Grant Disclosures

The following grant information was disclosed by the authors:
Slovenian Research Agency (ARRS): 36377.

### Competing Interests

The authors declare there are no competing interests.

### Author Contributions

- Miha Tome and Jure Zupan conceived and designed the experiments, performed the experiments, analyzed the data, contributed reagents/materials/analysis tools, prepared figures and/or tables, authored or reviewed drafts of the paper, approved the final draft.
- Zorica Tomičić conceived and designed the experiments, performed the experiments, analyzed the data, authored or reviewed drafts of the paper, approved the final draft.
- Tadeja Matos and Peter Raspor conceived and designed the experiments, contributed reagents/materials/analysis tools, authored or reviewed drafts of the paper, approved the final draft.

### Data Availablility

The raw data are provided in the Supplemental Files.

## Supplemental Information

Supplemental information for this article can be found online at http://dx.doi.org/10.7717/peerj.4999#supplemental-information.

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
