# Peer review of "Synergistic and antagonistic effects of immunomodulatory drugs on the action of antifungals against Candida glabrata and Saccharomyces cerevisiae"

_PeerJ, doi:10.7717/peerj.4999_

## Round 0.1 · original submission · Major Revisions

While this study presents some interesting data, as the reviewers both state the claims and scope are over-stated. In particular, antagonism seems to be quite strain-specific as it is not seen across all of the included strains (in fact some show additivity instead), and nothing is presented to determine why this might be the case or to follow it up in more strains. Both reviewers have significant other issues that need to be addressed before this paper can be accepted.

·

Basic reporting

This manuscript investigates drug interactions between antifungal and immunomodulatory drugs in C. glabrata and S. cerevisiae, in the context that these drugs can be co-administrated in a clinical setting. Overall the structure and concept of this manuscript is easy to read and comprehend. However, parts of this manuscript are unclear and need clarification in writing and further experimental procedures. Please see the following:
INTRODUCTION
• More background information is required for C. glabrata in the clinical setting. Because you focus on C. glabrata and not the major agent of candademia, more information is required to focus on why C. glabrata is important for this study. Are there differences between C. glabrata and C. albicans in terms of treatment, predominant population they affect?
• More context is needed for the mechanisms of drug resistance in Candida, especially to azoles as you focus on drug resistance in your manuscript
• It is ambiguous why S. cerevisiae was included in your study as you do not make further mention of S. cerevisiae past the checkerboard results. Will an antagonistic S. cerevisiae response to FLC+MPA be studied later and presented in another publication? Further justification of why S. cerevisiae was included is needed aside from that fact that it is a great model to understand the evolutionary basis it presents in understanding drug resistance development
• Please expand on the scope of drug combinations investigated by the scientific community. There’s plenty of literature that investigate drug and chemical library screens that synergise with antifungal drugs
• More background information is needed in terms of the types of immunomodulatory drugs that are encountered in the clinical setting
• More background information is required for MPA which becomes your main focus – is it known to have antifungal properties, what pathways does it target etc
• Most studies screen for synergistic drug interactions to understand the mechanism behind them. You focused specifically on antagonism for which your conclusions for doing so at the discussion and conclusion is sound. However, you need to justify why you are looking at antagonism in the introduction
• The end of your introduction need results and conclusions

Experimental design

A random selection of clinical C. glabrata along with pathogenic S. cerevisiae were tested. Antagonism was found prevalently between MPA and FLC, therefore MPA was further screened with other azoles. Because azole resistance is predominantly associated with increased expressed of efflux pumps, mostly Cdr1, the gene expression of this efflux pump as well as Pdr1 which regulates Cdr1 was investigated in MPA and FLC individually and when combined.
Both the CLSI based checkerboard assay and BI are accepted methods of investigating drug interactions. However, there are experimental methods that need clarification, more information and need to be conducted to a rigorous standard.
Below are the amendments needed:
• It would be beneficial to state the MICs of your selected C. glabrata and S. cerevisiae strains to drugs tested in Table 1. There are trends where organisms with high MICs show interactions with synergy compared to organisms with low MICs. Strain Sc6 is from sorghum beer, please make amendments that this strain is not a clinical isolate or justify why this strain was tested.
• Lines 116-118. Please state the stock concentrations of all drugs used.
• Lines 121-123. Please state the modifications made to the CLSI checkerboard assay - the CSLI protocol prepares all final drug concentrations in RPMI/test media and are performed in microtitre plates.
• Lines 125-126. Different cell concentrations were used for singular and combined drug susceptibility assays. While both 3x103 and 5x103 cells/mL both fall into the range needed for inoculating microtitre drug plates, is there a reason why these tests need different cell inocula? Justification is needed. Is the reference Zupan et al., 2013 used for the cell counting technique or a higher cell inoculum? If the latter, this reference is not suitable.
• Line 129. Is there a specific reason why assays were incubated at 37 °C when strains were grown at 35 °C? Usually assays are incubated at either 35 or 37 °C.
• Line 134. What is the genotype of the Cgpdr1Δ mutant?
• Line 134. Why was C. glabrata ATCC2001 not used in the initial drug screening and only for antagonism with azoles and immunomodulatory drugs? If you were to do further screening, it makes no sense to use a strain that was not in the initial drug screening, especially if the further screening only involves one strain. I would recommend testing C. glabrata ATCC2001 with the other antifungal + immunomodulatory drugs used in the first screen and incorporating these results in Table 1.
• Please state the concentration ranges tested for antifungal and immunomodulatory drugs
• Line 135. MICs change with media type. MICs tend to increase in rich media (YPD) compared to minimal media (RPMI). What was the reason the switch in media was made? Auxotrophy? If so, please substitute the required amino acids into RPMI and redo the MPA+FLC checkerboard with Cgpdr1Δ mutant.
• Line 144. Azoles have a trailing effect and their MICs can be read at either MIC50 or MIC80. Which one was used? Please state this for immunomodulatory drugs as well if a trailing effect is seen.
• Line 165. The concentrations used for FLC and MPA in the gene expression experiments may have been predetermined but they do not correspond to Figure 2A. Please supply the preliminary growth tests and checkerboard assay results are supplementary data.
• Line 168. Please state your reasoning behind choosing timepoints 2 and 4h? Are there literature showing the expression of drug efflux can be seen at 2 and 4 h in particular environments? If so, please reference them.
• Lines 181-182. Statistical analysis needs to be performed for the comparison of qPCR data, please include this. Also state the number of replicates used.

Validity of the findings

Reporting of findings were skeletal and lacking in robustness. Interpretation of results need to be defined for consistency for drug interactions. There are inconsistencies with what is reported and represented in the figures/tables. Please see the amendments:
• Line 187. Please state the hours of incubation as three incubation times were used.
• Line 188. Please state the range of concentrations strains exhibited resistance to FLC and ITR
• Line 191. State the strain and the MICs obtained
• Please reformat Table 2 so values and headings are legible in one line. State if MIC values are range, geomean or mode. State in footnotes MIC ranges that are considered susceptible, resistant etc. Alternatively reference this
• Lines 197-199. Please quantify drug interaction findings instead of stating a majority showed synergy etc. E.g., 1 out of 6 strains exhibited synergy with drug A and drug B.
• Lines 200-201. Please clarify ‘mixed responses’. Do you mean the interactions are indifferent? Or that across all strains tested, drug interactions were strain specific? All drug combinations are covered except for MPA+ITC. Please comment on this combination
• Lines 202-203. Please clarify ‘BI scored more hits’. Does this mean more synergy/antagonism was seen with BI vs FICI? Is there are good correlation between FICI and BI? Interpretation of drug interaction with these two models combined as you have done in Figure 1 needs to be defined. E.g., Cg3 AMX+FLC FICI show indifference, but BI shows synergy. Is this combination considered synergistic because at least one model is? This needs to be clarified for Figure 1 and in methods.
• Figure 1 needs a better key. A gradient in colours may be misread as level of drug interaction and not the level of consensus between FICI and BI. Perhaps separate out the colours as a separate key (you’re using a colour gradient) and the placements of FIC/SYN/ANT. Please state if FIC max or FIC min is used.
• Tables 3 does not state MIC results at RPMI, YPD, different temperatures or different time points as stated in text. Please amend this. Footnote of ‘n’ is ambiguous. Do you mean number of replicates? What is meant by ‘among 49 drug combinations for each strain’ when you only tested MPA + azole combinations with ATCC2001?
• Line 210. MPA MIC should be in a table format with RPMI YPD different temperatures tested etc
• Line 211. How did you assess timepoints as early as 4h? Did you assay this through back plating? If so, this needs to be stated in methods as to what protocol was being used and the results need to be shown. Please justify/reference the reasoning behind the choice of timepoints examined
• Line 212. Last sentence is too big a conclusion to draw, especially as these results are drawn from one strain alone. Restate this sentence.
• Line 222. Is there a statistically significant difference between gene expressions in the different treatment types? Incorporation of statistical significance is required
• Line 223. Rearrange sentence to make it clear that you are comparing between the two different timepoints
• Line 225. Rewrite paragraph so comparison of results do not jump across different parameters being measured
• Lines 229-238. As it was shown Pdr1 is not central to MPA+FLC antagonism, was Cdr1 confirmed to be central to this antagonism? qPCR results will be more robust with testing a Cgcdr1Δ mutant with MPA+FLC. Have you considered the expression of other drug efflux transporters?
• Figure 2. Please rearrange A B C D data in consecutive order they were discussed in the results. Antagonism was stated to be emphasised at MPA 8ug/ml and FLC 4 ug/ml. Why were qPCR experiments not performed at these drug concentrations? Please justify this lack of consistency in your methods
• Figure 2A. The interpretation of antagonism in the graphs given is not immediately clear. What are the cutoffs in growth for the MICs of both drugs tested? Is the interpretation of antagonism a 2 fold increase in MIC for FLC? If so, this does not follow FICI e.g., if The MIC of FLC alone is 64 ug/ml, having an MIC of 128 in combination with MPA at 16 ug/ml is not considered antagonistic. Please also state the media used for this test
• Figure 2B. Same with the comment above. The stated antagonistic combination of 4 ug/ml FLC + 8 ug/ml MPA is not considered antagonistic according to FICI. Please justify this. Better representative data is needed
• Line 235. Better to show BI interaction surface for Cgpdr1Δ mutant with MPA+FLC. Please comment on the decrease in FLC MIC seen here and not just in the discussion
DISCUSSION
More information is required for a robust discussion, specifically why you focused on the drug antagonism and the antagonism between MPA+FLC in the clinical setting. Additionally, the possibility of other drug efflux mechanisms that may play a role in MPA+FLC antagonism and antagonism from the MPA side of the story. Please also see the following:
• It is not clear why pathogenic S. cerevisiae strains were included in the initial screen between antifungal and immunomodulatory drugs as you make no mention of them in the discussion. Please clarify this.
• Line 245. Drug interactions can also be dangerous because of adverse reactions in the host. It is a more rounded argument to discuss this as well as the unwanted side effects of inducing drug resistance in a pathogen
• Line 248. Clarity is needed as to how FICI and BI correlated well – need to define this in the methods and interpretation of results. Are there literature that you can cite which talks about the correlation of FICI and BI to support your claims?
• Lines 255-257. You did not explore mechanisms of drug synergy is your study and therefore cannot make the claim that your ‘evaluation and interpretation of azoles and immunomodulatory drugs are on track’. Please rephrase this sentence. There are quite a lot of literature that report synergism between antifungal drugs and calcineurin inhibitors so reference these.
• Lines 263-264. Please clarify which drug the strains have a high resistance to
• Lines 264-265. Please clarify what the mechanisms of drug resistance may be
• Lines 262-268. This section does not sound like a discussion and you don’t talk about the validity of your findings here. Discuss about the potential consequences of MPA+FLC antagonism in the clinical setting here. What is the frequency of MPA+azole use in the clinics? Why would this combination be important?
• Line 272-274. Statistical analyses is required to support this claim
• Line 277 – 278. This can be further supported by qPCR-ing CDR1 in Cgpdr1Δ
• Lines 280-281. Clarification or a better representation of data is further needed for this method and the interpretation of these results as antagonistic
• Lines 286-287. You may also want to investigate antagonism from the MPA side of drug interactions, not just solely on the FLC side
• Lines 296-297. ‘We can speculate that this antagonism could be connected to this increase in fungal infections’ is an invalid claim. MPA is an immunosuppressant and the incidences of fungal infections as secondary complications are reviewed in Ritter & Priofski 2009. There is no claim to an increase in fungal infections with the use of MPA, let alone with MPA and azole drugs in combination. Additionally, there is no account of patients receiving antifungal and immunomodulatory drugs at the same time in the review and associated literature within. Please rephrase this sentence
• Lines 297-300. Please rephrase this sentence. Which drug combination is being referred to? What do you mean by ‘entire set of drugs for a prescribe treatment’?
CONCLUSION
I like the end of the conclusion and the statements presented mostly connect to the original questions investigated. Please find the following amendments and also make amendments according to those from the discussion section above:
• Line 306. ‘discovered an antagonistic interaction between MPA+azoles which is probably due to increased expression of drug efflux pump’. This sentence needs to be rephrased as you confirmed antagonism between MPA and azoles and performed qPCR experiments in only ATCC2001

Additional comments

Overall English language is legible. There are some sentences that are ambiguous and need clarification. Please see the following:
• The aims for the manuscript in the last paragraph need to be tidied up. Expanding on the mentioned points in the introduction will certainly help the aims/goals of this manuscript.
• Be consistent when referring to checkerboard/combination assays – choose one way to refer tests
• Line 24. Common to clinical environment meaning they are already in use or that they are common due to nature of diseases seen in clinics?
• Line 63. Resistance development and evolution – some resistance to drugs are transient
• Lines 60-62. Not just the side effects, the time and money taken to develop new drugs is problematic.
• Line 62. Synergistic and additive drug treatments not drug interactions
• Line 64. Be clear about functions of pathogen – mechanisms of pathogenesis?
• Line 65. Drug combinations currently used in clinics? Potential drug combinations?
• Line 68. Drug combinations in the clinical environment are overlooked?
• Line 75. What are these issues – does this belong to previous paragraph or existing paragraph? State issues first then state goals of project
• Line 77-78. Do you mean ‘To explore the modulatory effects of immunomodulatory and antifungal drugs in vitro’?
• Line 84. Please clarify this sentence so it corresponds to your methods
• Lines 94-95. Make it clear which strains are QC strains for drug testing and which one is used for antagonism studies
• Line 124. Suggest to state percentage of saline water, not 8.5g/L NaCl
• Line 144. Please reference Meletiadis et al., 2005 instead of Liu et al., 2014
• Line 162. Please change ‘Freshly sub-cultured C. glabrata ATCC2001 grew in liquid YPD overnight at 37 C’ to: C. glabrata ATCC2001 was grown in YPD overnight at 37 C
• Line190. Change ‘did no reach MIC’ to ‘did not obtain an MIC’
• Line 208. State what azole antifungals were tested
• Line 243. Drug interactions is not a practice. Please amend sentence
• Lines 246-247. Ambiguous sentence, needs to be clarified
• Line 259. Please clarify what ‘they’ is
• Line 283. Change exited to existed
• Line 287. Rephrase ‘Antagonism: Clinical application’ as antagonistic drug combinations will not be applied in clinical settings
• Line 296. Please cite original paper for increased fungal infections with MPA use

·

Basic reporting

English is sound, throughout the manuscript. However, the paragraph starting in line 185 should be completely re-written. It is an absurdly incomplete, superficial and poorly writen description of results. For example, the sentence starting in line 187 mentions "some strains" and "certain antifungals". Nothing could be more vague!

Minor points:
Table 2 - what is meant by cross-reference "a"? If nothing, remove it.
Table 3 - cross-reference "a" is not mentioned in this table?
I suggest that in both tables only the acronyms are described, with no need for "a" or "b".

Experimental design

The experiments were carefully planned and professionally executed. However, the following few issues should be addressed:

1 - The scope of the paper is on C. glabrata. The addition of S. cerevisiae can clearly be justified for its role as a model organism. But its mention as a pathogen should be very carefull. Please be less emphatic in sentences starting in lines 20 and 56.
2 - The introduction section should be more clear in terms of what is already known on the use of drug combinations. The authors mention "several drug combinations ..." (line 65). How do previous studies compare to yours? Have the same immunomodulatory drugs been tested before for combinatorial effect with antifungals?
3 - The selection of the 6 out of nearly 50 clinical isolates is not well justified in the paragraph starting in line 88. Authors should present the MICs for the whole collection to justify the selection of the 6. And given their selection, results should be analysed in light of the MIC exhibitted by each strain.
4 - Other Candida species mentioned in lines 95 and 96 were not used, were they? If not, discard them from the M&M section.
5 - line 101 - Strains were not preserved in just glycerol, were they? Please specify medium and % of glycerol.

Validity of the findings

There is a problem with the interpretation of the results in table 2. The major finding of this manuscript, that is the antagonism between MPA and FLC is not observed for all the tested strains. Indeed it is only seen for 3 out of 6 Cg strains, while for 2 of the remaining there seems to be a synergistic effect! A more clear discussion on these results must be included, and I suggest additional work in this point:

- given the proposed model that Cdr1 up-regulation is the reason for the antagonistic effect of both drugs, Cdr1 expression studies should be extended to the 6 Cg strains. By doing so the authors could establish a more clear correlation between Cdr1 responsiveness and the antagonistic effect, or absence of it in some strains. The fact that the effect might be strain dependent decreases the impact of this finding!

- please add statistical analysis to the results displayed in figures 2C and 2D.

- one of the interesting findings of this study is the suggestion that TFs other than Pdr1 may control CDR1 expression in the presence of MPA. Which TFs (line 237-8)? A few could be suggested based on the info gathered in the PathoYeastract database.

- the conclusion stated in line 250 that BI is more sensitive that FICI seems innapropriate. It would also be possible to argue that it is not more sensitive, it is more misleading, wouldn't it? If the authors have strong arguments for their point, please provide them.

Additional comments

Nothing to add.

---

## Round 0.2 · Minor Revisions

Thank you for substantially revising your manuscript, which is now greatly improved. Reviewer 1 has noted a number of areas that are still unclear that need further minor revision. Please ensure that all supplementary tables are clearly referenced in the text. Note that there is no editing for language or style undertaken by PeerJ and it is recommended that the revised manuscript be thoroughly proof-read by a native English speaker.

·

Basic reporting

Thank you for making the corrections, the manuscript has significantly improved. There are still a few things that need to be added and corrected. Please see the following suggestions:

Introduction

• Line 57. Does ‘a higher rate of antifungal use and the resistance of C. glabrata’ refer to innate resistance or developed resistance? Please clarify.
• Line 78. ‘activating mutations in the gene PDR1’. Do you mean mutations that result in the overactivation of this gene? Please clarify this sentence.
• Lines 81-83. Are all these genes in the genus Candida or specific to C. glabrata? Please clarify.
• Line 86. Please capitalise and italicise Erg11.
• Lines 88-91. Please clarify this sentence, it does not follow on from the previous sentence and the ending doesn’t make sense.
• Lines 94-98. Are you drawing similarities between S. cerevisiae and C. glabrata in the first sentence? Please clarify the first sentence or delete it as it is redundant.
• Lines 118-121. Please clarify that both AMB and 5FC are antifungal drugs. Is this combination used for Candida? Or is this combination used in general for mycoses?
• Line 124. What studies is being referred to here?
• Line 125. What inhibitor is being referred to? Of a protein of interest?
• Line 142.Does MTX have the same antifungal mode of action as calcineurin? If not, please amend sentence.
• Line 145. Does mycophenolic acid have an abbreviation?
• Line 160. Please change ‘usually’ to commonly.

Experimental design

Please see the following suggestions:

• Line 230. ‘200 to 2 mg/l’. Please make this consistent with how the other ranges are written.
• Line 236. Please clarify if the triplicates are biological or technical.

Validity of the findings

Results:

• Line 296. Please clarify ‘expected range’. Do you mean of all the antifungals tested?
• Line 300. Please reference CLSI.
• Line 314. How many S. cerevisiae strains and C. glabrata strains?
• Line 316. What is the measure of uniformity here? 8/12 strains? Was it the same S. cerevisiae and C. glabrata strains in the 8/12 strains?
• Lines 318-322. Are there any strains that showed synergism/antagonism across MTX or AMX?
• Line 324. What are Cg5 and Cg6 resistant to?
• Line 353. Please replace ‘two genes’ with PDR1 and CDR1.
• Line 353. Please state how you did this in the methods. How did you normalise your data?


Discussion and conclusions

• Line 386. Drug combinations in clinical settings are often unintentional and just occur and not achieved. Please amend sentence.
• Line 390-391. Are these 8/12 strains the same strains that were antagonistic in FLC+MPA and synergistic in AMB+MPA? If not, please clarify.
• Line 411-413. Please clarify sentence, it is not clear what you want to convey here.
• Line 420. Please clarify what is meant by ‘this effect’.
• Lines 421-423. This sentence makes it sound as though efflux pumps are not studied in S. cerevisiae or C. glabrata. Please amend this sentence.
• Line 430. Please state increased resistance to what.
• Line 435. Please state what the combination is for clarity.
• Line 463. Please change ‘tolerance’ to resistance.

Additional comments

Previous suggestions and responses. Please see my responses starting with '>':

• It would be beneficial to state the MICs of your selected C. glabrata and S. cerevisiae strains to drugs tested in Table 1. There are trends where organisms with high MICs show interactions with synergy compared to organisms with low MICs. Strain Sc6 is from sorghum beer, please make amendments that this strain is not a clinical isolate or justify why this strain was tested.
We agree, the MICs would be a good addition to the Table 1. However, in our opinion the addition would make the Table 1 too crowded and unclear. This is the reason we made a separate Table 2, where we have strains with their MICs. Additionally, we added a list of all the tested strains for the drug susceptibility (the ones mentioned in lines 90–91) to Supplemental Files (“List_of_strain.xlsx”).

We have added an explanation, why we picked Sc6 (from sorghum beer; it displayed high tolerance to fluconazole for a non-clinical isolate) and corrected it throughout the text (mostly we replaced “clinical isolates”, since they are only 11 out of 12, with “selected strains”).

> It is great that you have provided the MICs of the strains you have used. Please refer this data to the supplementary file in your manuscript so readers can refer to it if they wish to.


• Lines 116-118. Please state the stock concentrations of all drugs used.

We can add the stock concentrations in the Supplement Files. However, we think that they aren’t as relevant to the publication and would distract from the clarity. Each experiment had a different set of stocks, the stock solution was usually prepared for each experiment individually depending on the final concentration of drugs, keeping in mind, that the relative DMSO value at the end concentration was always below 1 %. Publications with similar methodology (for example (Homa et al., 2016; Moreno-Martinez et al., 2015; Reichert-Lima et al., 2016; Valentín et al., 2016)) usually only have stated test ranges for each drug (we added this), but the stock solution concentrations are left out, since the data doesn’t add anything really scientifically important.

> Please put these in the supplementary files. It is for the benefit of other who want to replicate your methods from start to finish.


• Lines 125-126. Different cell concentrations were used for singular and combined drug susceptibility assays. While both 3x103 and 5x103 cells/mL both fall into the range needed for inoculating microtitre drug plates, is there a reason why these tests need different cell inocula? Justification is needed. Is the reference Zupan et al., 2013 used for the cell counting technique or a higher cell inoculum? If the latter, this reference is not suitable.

There wasn’t a particular reason for the different concentrations, both were inside the range defined by the CLSI checkerboard assay. The susceptibility to a single drug was performed first and in later for the checkerboard assays, we used the slightly higher concentration. We haven’t notice this to have any effect and to be completely frank, we aren’t really sure why this change happened. The focus was just to stay inside the prescribed range. We also argue, that this difference doesn’t have an impact.
We could maybe write it in a less confusing way, similar as in (Reichert-Lima et al., 2016): “The inoculums were 0.5 x 103 to 2.5 * 103 CFU ml-1”. Or in our case something along the line: “Inoculum was prepared by diluting yeast cells in RPMI medium to 3 x 103 to 5 x 103 cells/ml using the automatic ImageJ-counting technique...”.
The Zupan et al., 2013 reference is used for the cell counting technique.

> It is true it will not make a difference, but to not distract from clarity, please write it according to Reichert-Lima et al.


• Line 135. MICs change with media type. MICs tend to increase in rich media (YPD) compared to minimal media (RPMI). What was the reason the switch in media was made? Auxotrophy? If so, please substitute the required amino acids into RPMI and redo the MPA+FLC checkerboard with Cgpdr1Δ mutant.

We confirmed that the antagonistic effect is present in both media – results were added in the Supplement Files (“ATCC_2001_RPMI_YPD.xlsx”), and this was respected for further experiments (especially for the future studies of the mechanism via chemogenomics/transcriptomics). The MICs in YPD after 48 h were higher than in RPMI, but the induced tolerance was still present at the boarder concentrations of fluconazole.

> Please state that you have done this in RPMI as well as reference to it as supplementary material.


• Line 144. Azoles have a trailing effect and their MICs can be read at either MIC50 or MIC80. Which one was used? Please state this for immunomodulatory drugs as well if a trailing effect is seen.

MIC50 for azoles and immunomodulatory drugs, and MIC90 for amphotericin B. Added in the text.

> Why MIC90 for AMB? CLSI uses MIC of 100 for AMB.


• Line 165. The concentrations used for FLC and MPA in the gene expression experiments may have been predetermined but they do not correspond to Figure 2A. Please supply the preliminary growth tests and checkerboard assay results are supplementary data.

Added a Supplement file (“Growth_curve_time_points_for_gene_expression.docx”), which includes the growth curve and the references for the time points.

> Please refer to this in manuscript text.


• Please reformat Table 2 so values and headings are legible in one line. State if MIC values are range, geomean or mode. State in footnotes MIC ranges that are considered susceptible, resistant etc. Alternatively reference this.

Reformatted that the headings are in one line. It already states that the MIC values are range (in the heading), if this isn’t what you meant, we will require further explanation. In the footnote, the MIC ranges have the interpretation with added reference for the breakpoints. We haven’t changed how we marked the resistant strains. If this approach isn’t appropriate, please let us know.

> All good, no further clarification is needed.


• Line 222. Is there a statistically significant difference between gene expressions in the different treatment types? Incorporation of statistical significance is required

Added the analysis (in methods, changed figures 2C, 2D), corrected in the results.

> Please add p values in figures and manuscript text so readers know which comparisons show the significance of difference between gene expressions.
Another thing you must include for qPCR is the normalization of gene expression data with your house keeping genes. Please include in your method for this.


• Line 124. Suggest to state percentage of saline water, not 8.5g/L NaCl

Corrected.

> This reads as 8.5 g/L salt. Please change NaCl to saline solution.

·

Basic reporting

All the previously raised issues were adequately addressed.

Experimental design

All the previously raised issues were adequately addressed.

Validity of the findings

All the previously raised issues were adequately addressed.

Additional comments

All the previously raised issues were adequately addressed.

---

## Round 0.3 · Minor Revisions

Thank you for addressing the concerns of the reviewer. As PeerJ does not perform editing I have done some minor editing of your manuscript to improve style and to clarify any confusing or cumbersome sentences. A marked up version of your paper is attached. Please take a look at the changes and accept or amend your paper as advised.

I've asked the PeerJ staff about naming of your supplementary files and they should be in contact with you about this.

---

## Round 0.4 · accepted · Accept

Thank you for making all the necessary changes, and congratulations on a nice manuscript.

#